# Insights on representational similarity in neural networks with canonical correlation

**Ari S. Morcos**[*‡]
DeepMind[†]
arimorcos@gmail.com

**Maithra Raghu**[*‡]
Google Brain, Cornell University
maithrar@gmail.com

**Samy Bengio**
Google Brain
bengio@google.com

## Abstract

Comparing different neural network representations and determining how representations evolve over time remain challenging open questions in our understanding of the function of neural networks. Comparing representations in neural networks is fundamentally difficult as the structure of representations varies greatly, even across groups of networks trained on identical tasks, and over the course of training. Here, we develop projection weighted CCA (Canonical Correlation Analysis) as a tool for understanding neural networks, building off of SVCCA, a recently proposed method [22]. We first improve the core method, showing how to differentiate between signal and noise, and then apply this technique to compare across a group of CNNs, demonstrating that networks which generalize converge to more similar representations than networks which memorize, that wider networks converge to more similar solutions than narrow networks, and that trained networks with identical topology but different learning rates converge to distinct clusters with diverse representations. We also investigate the representational dynamics of RNNs, across both training and sequential timesteps, finding that RNNs converge in a bottom-up pattern over the course of training and that the hidden state is highly variable over the course of a sequence, even when accounting for linear transforms. Together, these results provide new insights into the function of CNNs and RNNs, and demonstrate the utility of using CCA to understand representations.

## 1 Introduction

As neural networks have become more powerful, an increasing number of studies have sought to decipher their internal representations [26, 16, 4, 2, 11, 25, 21]. Most of these have focused on the role of individual units in the computations performed by individual networks. Comparing population representations across networks has proven especially difficult, largely because networks converge to apparently distinct solutions in which it is difficult to find one-to-one mappings of units [16].

Recently, [22] applied Canonical Correlation Analysis (CCA) as a tool to compare representations across networks. CCA had previously been used for tasks such as computing the similarity between modeled and measured brain activity [23], and training multi-lingual word embeddings in language models [5]. Because CCA is invariant to linear transforms, it is capable of finding shared structure across representations which are superficially dissimilar, making CCA an ideal tool for comparing the representations across groups of networks and for comparing representations across time in RNNs.

Using CCA to investigate the representations of neural networks, we make three main contributions:

---

[*]equal contribution, in alphabetical order

[†]Work done while at DeepMind; currently at Facebook AI Research (FAIR)

[‡]To whom correspondence should be addressed: arimorcos@gmail.com, maithrar@gmail.com

1. We analyse the technique introduced in [22], and identify a key challenge: the method does not effectively distinguish between the signal and the noise in the representation. We address this via a better aggregation technique (Section 2.2).

2. Building off of [21], we demonstrate that groups of networks which generalize converge to more similar solutions than those which memorize (Section 3.1), that wider networks converge to more similar solutions than narrower networks (Section 3.2), and that networks with identical topology but distinct learning rates converge to a small set of diverse solutions (Section 3.3).

3. Using CCA to analyze RNN representations over training, we find that, as with CNNs [22], RNNs exhibit bottom-up convergence (Section 4.1). Across sequence timesteps, however, we find that RNN representations vary significantly (Section A.3).

## 2 Canonical Correlation Analysis on Neural Network Representations

Canonical Correlation Analysis [10], is a statistical technique for relating two sets of observations arising from an underlying process. It identifies the 'best' (maximizing correlation) linear relationships (under mutual orthogonality and norm constraints) between two sets of multidimensional variates.

Concretely, in our setting, the underlying process is a neural network being trained on some task. The multidimensional variates are *neuron activation vectors* over some dataset $X$. As in [22], a neuron activation vector denotes the outputs a single neuron $z$ has on $X$. If $X = \{x_1, ..., x_m\}$, then the neuron $z$ outputs scalars $z(x_1), ..., z(x_m)$, which can be stacked to form a vector.[4]

A single neuron activation vector is one multidimensional variate, and a layer of neurons gives us a *set* of multidimensional variates. In particular, we can consider two layers, $L_1$, $L_2$ of a neural network as two sets of observations, to which we can then apply CCA, to determine the similarity between two layers. Crucially, this similarity measure is invariant to (invertible) affine transforms of either layer, which makes it especially apt for neural networks, where the representation at each layer typically goes through an affine transform before later use. Most importantly, it also enables comparisons between *different* neural networks,[5] which is not naively possible due to a lack of any kind of neuron to neuron alignment.

### 2.1 Mathematical Details of Canonical Correlation

Here we overview the formal mathematical interpretation of CCA, as well as the optimization problem to compute it. Let $L_1, L_2$ be $a \times n$ and $b \times n$ dimensional matrices respectively, with $L_1$ representing $a$ multidimensional variates, and $L_2$ representing $b$ multidimensional variates. We wish to find vectors $w, s$ in $\mathbb{R}^a, \mathbb{R}^b$ respectively, such that the dot product

$$\rho = \frac{\langle w^T L_1, s^T L_2 \rangle}{||w^T L_1|| \cdot ||s^T L_2||}$$

is maximized. Assuming the variates in $L_1, L_2$ are centered, and letting $\Sigma_{L_1,L_1}$ denote the $a$ by $a$ covariance of $L_1$, $\Sigma_{L_2,L_2}$ denote the $b$ by $b$ covariance of $L_2$, and $\Sigma_{L_1,L_2}$ the cross covariance:

$$\frac{\langle w^T L_1, s^T L_2 \rangle}{||w^T L_1|| \cdot ||s^T L_2||} = \frac{w^T \Sigma_{L_1,L_2} s}{\sqrt{w^T \Sigma_{L_1,L_1} w} \sqrt{s^T \Sigma_{L_2,L_2} s}}$$

We can change basis, to $w = \Sigma_{L_1,L_1}^{-1/2} u$ and $s = \Sigma_{L_2,L_2}^{-1/2} v$ to get

$$\frac{w^T \Sigma_{L_1,L_2} s}{\sqrt{w^T \Sigma_{L_1,L_1} w} \sqrt{s^T \Sigma_{L_2,L_2} s}} = \frac{u^T \Sigma_{L_1,L_1}^{-1/2} \Sigma_{L_1,L_2} \Sigma_{L_2,L_2}^{-1/2} v}{\sqrt{u^T u} \sqrt{v^T v}} \qquad (*)$$

which can be solved with a singular value decomposition:

$$\Sigma_{L_1,L_1}^{-1/2} \Sigma_{L_1,L_2} \Sigma_{L_2,L_2}^{-1/2} = U \Lambda V$$

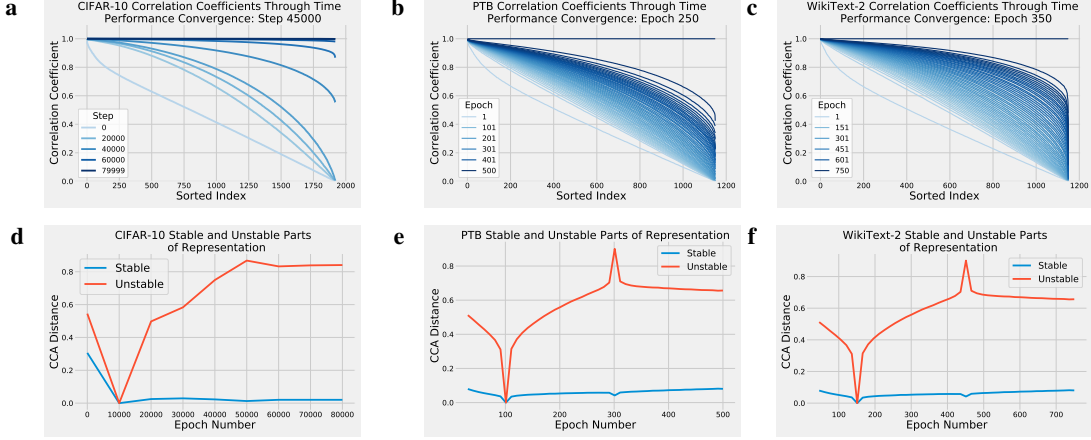

**Figure 1: CCA distinguishes between stable and unstable parts of the representation over the course of training.** Sorted CCA coefficients $(\rho_t^{(i)})$ comparing representations between layer $L$ at times $t$ through training with its representation at the final timestep $T$ for CNNs trained on CIFAR-10 (**a**), and RNNs trained on PTB (**b**) and WikiText-2 (**c**). For all of these networks, at time $t_0 < T$ (indicated in title), the performance converges to match final performance (see Figure A1). However, many $\rho_t^{(i)}$ are unconverged, corresponding to unnecessary parts of the representation (noise). To distinguish between the signal and noise portions of the representation, we apply CCA between $L$ at timestep $t_{early}$ early in training, and $L$ at timestep $T/2$ to get $\rho_{T/2}$. We take the 100 top converged vectors (according to $\rho_{T/2}$) to form $S$, and the 100 least converged vectors to form $B$. We then compute CCA similarity between $S$ and $L$ at time $t > t_{early}$, and similarly for $B$. $S$ remains stable through training (signal), while $B$ rapidly becomes uncorrelated (**d-f**). Note that the sudden spike at $T/2$ in the unstable representation is because it is chosen to be the least correlated with step $T/2$.

with $u, v$ in (*) being the first left and right singular vectors, and the top singular value of $\Lambda$ corresponding to the canonical correlation coefficient $\rho \in [0, 1]$, which tells us how well correlated the vectors $w^T L_1 = u^T \Sigma_{L_1, L_1}^{-1/2} L_1$ and $s^T L_2 = v^T \Sigma_{L_2, L_2}^{-1/2} L_2$ (both vectors in $\mathbb{R}^n$) are.

In fact, $u, v, \rho$ are really the first in a series, and can be denoted $u^{(1)}, v^{(1)}, \rho^{(1)}$. Next in the series are $u^{(2)}, v^{(2)}$, the second left and right singular vectors, and $\rho^{(2)}$ the corresponding second highest singular value of $\Lambda$. $\rho^{(2)}$ denotes the correlation between $(u^{(2)})^T \Sigma_{L_1, L_1}^{-1/2} L_1$ and $(v^{(2)})^T \Sigma_{L_2, L_2}^{-1/2} L_2$, which is the next highest possible correlation under the constraint that $\langle u^{(1)}, u^{(2)} \rangle = 0$ and $\langle v^{(1)}, v^{(2)} \rangle = 0$.

The output of CCA is a series of singular vectors $u^{(i)}, v^{(i)}$ which are pairwise orthogonal, their corresponding vectors in $\mathbb{R}^n$: $(u^{(i)})^T \Sigma_{L_1, L_1}^{-1/2} L_1$ and $(v^{(i)})^T \Sigma_{L_2, L_2}^{-1/2} L_2$, and finally their correlation coefficient $\rho^{(i)} \in [0, 1]$, with $\rho^{(i)} \le \rho^{(j)}, i > j$. Letting $c = \min(a, b)$, we end up with $c$ non-zero $\rho^{(i)}$.

Note that the orthogonality of $u^{(i)}, u^{(j)}$ also results in the orthogonality of $(u^{(i)})^T \Sigma_{L_1, L_1}^{-1/2} L_1, (u^{(j)})^T \Sigma_{L_1, L_1}^{-1/2} L_1$, as

$$\langle (u^{(i)})^T \Sigma_{L_1, L_1}^{-1/2} L_1, (u^{(j)})^T \Sigma_{L_1, L_1}^{-1/2} L_1 \rangle = (u^{(i)})^T \Sigma_{L_1, L_1}^{-1/2} L_1 L_1^T \Sigma_{L_1, L_1}^{-1/2} (u^{(j)}) = (u^{(i)})^T (u^{(j)}) = 0 \quad (**)$$

and so our CCA directions are also orthogonal.

## 2.2 Beyond Mean CCA Similarity

To determine the representational similarity between two layers $L_1, L_2$, [22] prunes neurons with a preprocessing SVD step, and then applies CCA to $L_1, L_2$. They then represent the similarity of $L_1, L_2$ by the *mean* correlation coefficient. Adapting this to make a distance measure, $d_{SVCCA}(L_1, L_2)$:

$$d_{SVCCA}(L_1, L_2) = 1 - \frac{1}{c} \sum_{i=1}^{c} \rho^{(i)}$$

One drawback with this measure is that it implicitly assumes that all $c$ CCA vectors are equally important to the representations at layer $L_1$. However, there has been ample evidence that DNNs do

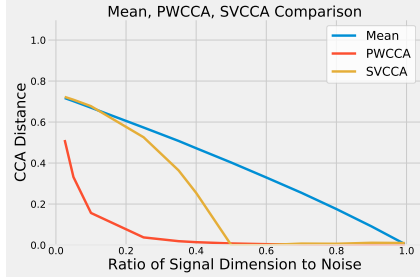

**Figure 2: Projection weighted (PWCCA) vs. SVCCA vs. unweighted mean** Unweighted mean (blue) and projection weighted mean (red) were used to compare synthetic ground truth signal and uncommon (noise) structure, each of fixed dimensionality. As the signal to noise ratio decreases, the unweighted mean underestimates the shared structure, while the projection weighted mean remains largely robust. SVCCA performs better than the unweighted mean but less well than the projection weighting.

not rely on the full dimensionality of a layer to represent high performance solutions [12, 6, 1, 20, 15, 21, 14]. As a result, the mean correlation coefficient will typically underestimate the degree of similarity.

To investigate this further, we first asked whether, over the course of training, all CCA vectors converge to their final representations before the network's performance converges. To test this, we computed the CCA similarity between layer $L$ at times $t$ throughout training with layer $L$ at the final timestep $T$. Viewing the sorted CCA coefficients $\rho$, we can see that many of the coefficients continue to change well after the network's performance has converged (Figure 1a-c, Figure A1). This result suggests that the unconverged coefficients and their corresponding vectors may represent "noise" which is unnecessary for high network performance.

We next asked whether the CCA vectors which stabilize early in training remain stable. To test this, we computed the CCA vectors between layer $L$ at timestep $t_{early}$ in training and timestep $T/2$. We then computed the similarity between the top 100 vectors (those which stabilized early) and the bottom 100 vectors (those which had not stabilized) with the representation at all other training times. Consistent with our intuition, we found that those vectors which stabilized early remained stable, while the unstable vectors continued to vary, and therefore likely represent noise.

These results suggest that task-critical representations are learned by midway through training, while the noise only approaches its final value towards the end. We therefore suggest a simple and easy to compute variation that takes this into account. We also discuss an alternate approach in Section A.2.

**Projection Weighting**   One way to address this issue is to replace the mean by a weighted mean, in which canonical correlations which are more important to the underlying representation have higher weight. We propose a simple method, *projection weighting*, to determine these weights. We base our proposition on the hypothesis that CCA vectors that account for (loosely speaking) a larger proportion of the original outputs are likely to be more important to the underlying representation.

More formally, let layer $L_1$, have neuron activation vectors $[z_1, ..., z_a]$, and CCA vectors $h_i = (u^{(i)})^T \Sigma_{L_1, L_1}^{-1/2} L_1$. We know from (**) that $h_i, h_j$ are orthogonal. Because computing CCA can result in the accrual of small numerical errors [24], we first explicitly orthonormalize $h_1, ..., h_c$ via Gram-Schmidt. We then identify how much of the original output is accounted for by each $h_i$:

$$\tilde{\alpha}_i = \sum_j |\langle h_i, z_j \rangle|$$

Normalizing this to get weights $\alpha_i$, with $\sum_i \alpha_i = 1$, we can compute the projection weighted CCA distance[6]:

$$d(L_1, L_2) = 1 - \sum_{i=1}^{c} \alpha_i \rho^{(i)}$$

As a simple test of the benefits of projection weighting, we constructed a toy case in which we used CCA to compare the representations of two networks with common (signal) and uncommon (noise)

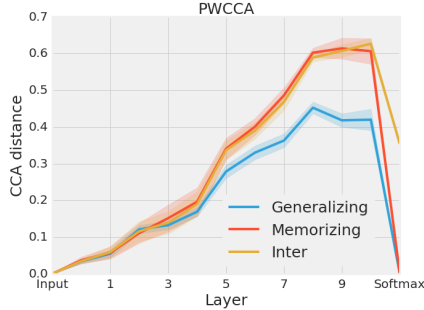

**Figure 3: Generalizing networks converge to more similar solutions than memorizing networks.** Groups of 5 networks were trained on CIFAR-10 with either true labels (generalizing) or a fixed random permutation of the labels (memorizing). The pairwise CCA distance was then compared within each group and between generalizing and memorizing networks (inter) for each layer, based on the training data, and the projection weighted CCA coefficient (with thresholding to remove low variance noise.) While both categories converged to similar solutions in early layers, likely reflecting convergent edge detectors, etc., generalizing networks converge to significantly more similar solutions in later layers. At the softmax, sets of both generalizing and memorizing networks converged to nearly identical solutions, as all networks achieved near-zero training loss. Error bars represent mean $\pm$ std weighted mean CCA distance across pairwise comparisons.

structure, each of a fixed dimensionality. We then used the naive mean and projected weighted mean to measure the CCA distance between these two networks as a function of the ratio of signal dimensions to noise dimensions. As expected we found that while the naive mean was extremely sensitive to this ratio, the projection weighted mean was largely robust (Figure 2).

## 3   Using CCA to measure the similarity of converged solutions

Because CCA measures the distance between two representations independent of linear transforms, it enables formerly difficult comparisons between the representations of different networks. Here, we use this property of CCA to evaluate whether groups of networks trained on CIFAR-10 with different random initializations converge to similar solutions under the following conditions:

- When trained on identically randomized labels (as in [27]) or on the true labels (Section 3.1)
- As network width is varied (Section 3.2)
- In a large sweep of 200 networks (Section 3.3)

### 3.1   Generalizing networks converge to more similar solutions than memorizing networks

It has recently been observed that DNNs are capable of solving image classification tasks even when the labels have been randomly permuted [27]. Such networks must, by definition, memorize the training data, and therefore cannot generalize beyond the training set. However, the representational properties which distinguish networks which memorize from those which generalize remain unclear.

In particular, we hypothesize that the representational similarity in a group of generalizing networks (networks trained on the true labels) should differ from the representational similarity of memorizing networks (networks trained on random labels.)

To test this hypothesis, we trained groups of five networks with identical topology on either unmodified CIFAR-10 or CIFAR-10 with random labels (the same set of random labels was used for all networks), all of which were trained to near-zero training loss[7]. Critically, the randomization of CIFAR-10 labels was consistent for all networks. To evaluate the similarity of converged solutions, we then measured the pairwise projection weighted CCA distance for each layer among networks trained on unmodified CIFAR-10 ("Generalizing"), among networks trained on randomized label CIFAR-10 ("Memorizing") and between each pair of networks trained on unmodified and random

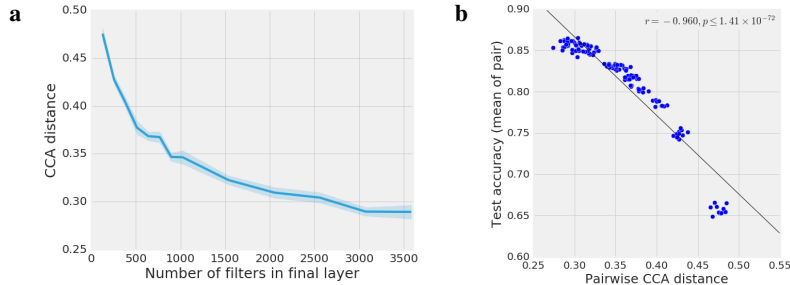

**Figure 4: Larger networks converge to more similar solutions.** Groups of 5 networks with different random initializations were trained on CIFAR-10. Pairwise CCA distance was computed for members of each group. Groups of larger networks converged to more similar solutions than groups of smaller networks (**a**). Test accuracy was highly correlated with degree of convergent similarity, as measured by CCA distance (**b**).

label CIFAR-10 ("Inter"). For all analyses, the representation in a given layer was obtained by averaging across all spatial locations within each filter.

Remarkably, we found that not only do generalizing networks converge to more similar solutions than memorizing networks (to be expected, since generalizing networks are more constrained), but memorizing networks are as similar to each other as they are to a generalizing network. This result suggests that the solutions found by memorizing networks were as diverse as those found across entirely different dataset labellings.

We also found that at early layers, all networks converged to equally similar solutions, regardless of whether they generalize or memorize (Figure 3). Intuitively, this makes sense as the feature detectors found in early layers of CNNs are likely required regardless of the dataset labelling. In contrast, however, at later layers, groups of generalizing networks converged to substantially more similar solutions than groups of memorizing networks (Figure 3). Even among networks which generalize, the CCA distance between solutions found in later layers was well above zero, suggesting that the solutions found were quite diverse. At the softmax layer, sets of both generalizing and memorizing networks converged to highly similar solutions when CCA distance was computed based on training data; when test data was used, however, only generalizing networks converged to similar softmax outputs (Figure A10), again reflecting that each memorizing network memorizes the training data using a different strategy.

Importantly, because each network learned a different linear transform of a similar solution, traditional distance metrics, such as cosine or Euclidean distance, were insufficient to reveal this difference (Figure A5). Additionally, while unweighted CCA revealed the same broad pattern, it does not reveal that generalizing networks get more similar in the final two layers (Figure A9).

## 3.2 Wider networks converge to more similar solutions

In the model compression literature, it has been repeatedly noted that while networks are robust to the removal of a large fraction of their parameters (in some cases, as many as 90%), networks initialized and trained from the start with fewer parameters converge to poorer solutions than those derived from pruning a large networks [8, 9, 6, 1, 20, 15]. Recently, [7] proposed the "lottery ticket hypothesis," which hypothesizes that larger networks are more likely to converge to good solutions because they are more likely to contain a sub-network with a "lucky" initialization. If this were true, we might expect that groups of larger networks are more likely to contain the same "lottery ticket" sub-network and are therefore more likely to converge to similar solutions than smaller networks.

To test this intuition, we trained groups of convolutional networks with increasing numbers of filters at each layer. We then used projection weighted CCA to measure the pairwise similarity between each group of networks of the same size. Consistent with our intuition, we found that larger networks converged to much more similar solutions than smaller networks (Figure 4a).[8] This is also consistent with the equivalence of deep networks to Gaussian processes (GPs) in the limit of infinite width

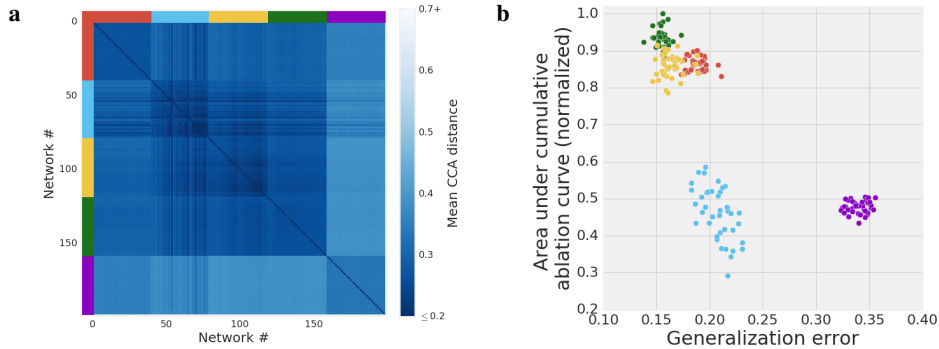

**Figure 5: CCA reveals clusters of converged solutions across networks with different random initializations and learning rates.** 200 networks with identical topology and varying learning rates were trained on CIFAR-10. CCA distance between the eighth layer of each pair of networks was computed, revealing five distinct subgroups of networks (**a**). These five subgroups align almost perfectly with the subgroups discovered in [21] (**b**; colors correspond to bars in **a**), despite the fact that the clusters in [21] were generated using robustness to cumulative ablation, an entirely separate metric.

[13, 17]. If each unit in a layer corresponds to a draw from a GP, then as the number of units increases the CCA distance will go to zero.

Interestingly, we also found that networks which converged to more similar solutions also achieved noticeably higher test accuracy. In fact, we found that across pairs of networks, the correlation between test accuracy and the pairwise CCA distance was -0.96 (Figure 4b), suggesting that the CCA distance between groups of identical networks with different random initializations (computed using the *train* data) may serve as *a strong predictor of test accuracy*. It may therefore enable accurate prediction of test performance without requiring the use of a validation set.

### 3.3 Across many initializations and learning rates, networks converge to discriminable clusters of solutions

Here, we ask whether networks trained on the same data with different initializations and learning rates converge to the same solutions. To test this, we measured the pairwise CCA distance between networks trained on unmodified CIFAR-10. Interestingly, when we plotted the pairwise distance matrix (Figure 5a), we observed a block diagonal structure consistent with five clusters of converged network solutions, with one cluster highly dissimilar to the other four clusters. Despite the fact that these networks all achieved similar train loss (and many reached similar test accuracy as well), these clusters corresponded with the learning rate used to train each network. This result suggests that there exist multiple minima in the optimization landscape to which networks may converge which are largely specified by the optimization parameters.

In [21], the authors also observed clusters of network solutions using the relationship between networks' robustness to cumulative deletion or "ablation" of filters and generalization error. To test whether the same clusters are found via these distinct approaches, we assigned a color to each cluster found using CCA (see bars on left and top in Figure 5a), and used these colors to identify the same networks in a plot of ablation robustness vs. generalization error (Figure 5b). Surprisingly, the clusters found using CCA aligned nearly perfectly with those observed using ablation robustness.

This result suggests not only that networks with different learning rates converge to distinct clusters of solutions, but also that these clusters can be uncovered independently using multiple methods, each of which measures a different property of the learned solution. Moreover, analyzing these networks using traditional metrics, such as generalization error, would obscure the differences between many of these networks.

---

anything, lead to an overestimate of the distance between groups of larger networks, as they are more heavily subsampled.

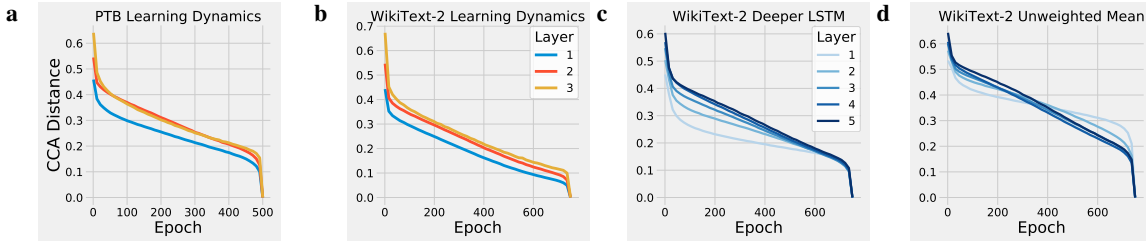

**Figure 6: RNNs exhibit bottom-up learning dynamics.** To test whether layers converge to their final representation over the course of training with a particular structure, we compared each layer's representation over the course of training to its final representation using CCA. In shallow RNNs trained on PTB (**a**), and WikiText-2 (**b**), we observed a clear bottom-up convergence pattern, in which early layers converge to their final representation before later layers. In deeper RNNs trained on WikiText-2, we observed a similar pattern (**c**). Importantly, the weighted mean reveals this effect much more accurately than the unweighted mean, which is also supported by control experiments (Figure A8) (**d**), revealing the importance of appropriate weighting of CCA coefficients.

# 4    CCA on Recurrent Neural Networks

So far, CCA has been used to study *feedforward* networks. We now use CCA to investigate RNNs. Our RNNs are LSTMs used for the Penn Treebank (PTB) and WikiText-2 (WT2) language modelling tasks, following the implementation in [18, 19].

One specific question we explore is whether the learning dynamics of RNNs mirror the "bottom up" convergence observed in the feedforward case in [22], as well as investigating whether CCA produces qualitatively better outputs than other metrics. However, in the case of RNNs, there are two possible notions of "time". There is the training timestep, which affects the values of the weights, but also a 'sequence timestep' – the number of tokens of the sequence that have been fed into the recurrent net. This latter notion of time does not explicitly change the weights, but results in updated values of the cell state and hidden state of the network, which of course affect the representations of the network.

In this work, we primarily focus on the training notion of time; however, we perform a preliminary investigation of the sequence notion of time as well, demonstrating that CCA is capable of finding similarity across sequence timesteps which are missed by traditional metrics (Figures A2, A4), but also that even CCA often fails to find similarity in the hidden state across sequence timesteps, suggesting that representations over sequence timesteps are often not linearly similar (Figure A3).

## 4.1    Learning Dynamics Through Training Time

To measure the convergence of representations through training time, we computed the projection weighted mean CCA value for each layer's representation throughout training to its final representation. We observed bottom-up convergence in both Penn Treebank and WikiText-2 (Figure 6a-b). We repeated these experiments with cosine and Euclidean distance (Figure A8), finding that while these other metrics also reveal a bottom up convergence, the results with CCA highlight this phenomena much more clearly.

We also observed bottom-up convergence in a deeper LSTM trained on WikiText-2 (the larger dataset) (Figure 6c). Interestingly, we found that this result changes noticeably if we use the unweighted mean CCA instead, demonstrating the importance of the weighting scheme (Figure 6d).

# 5    Discussion and future work

In this study, we developed CCA as a tool to gain insights on many representational properties of deep neural networks. We found that the representations in hidden layers of a neural network contain both "signal" components, which are stable over training and correspond to performance curves, and an unstable "noise" component. Using this insight, we proposed projection weighted CCA, adapting [22]. Leveraging the ability of CCA to compare across different networks, we investigated the properties of converged solutions of convolutional neural networks (Section 3), finding that

networks which generalize converge to more similar solutions than those which memorize (Section 3.1), that wider networks converge to more similar solutions than narrow networks (Section 3.2), and that across otherwise identical networks with different random initializations and learning rates, networks converge to diverse clusters of solutions (Section 3.3). We also used projection weighted CCA to study the dynamics (both across training time and sequence steps) of RNNs, (Section 4), finding that RNNs exhibit bottom-up convergence over the course of training (Section 4.1), and that across sequence timesteps, RNN representations vary nonlinearly (Section A.3).

One interesting direction for future work is to examine what is unique about directions which are preserved across networks trained with different initializations. Previous work has demonstrated that these directions are sufficient for the network computation [22], but the properties that make these directions special remain unknown. Furthermore, the attributes which specifically distinguish the diverse solutions found in Figure 5 remain unclear. We also observed that networks which converge to similar solutions exhibit higher generalization performance (Figure 4b). In future work, it would be interesting to explore whether this insight could be used as a regularizer to improve network performance. Additionally, it would be useful to explore whether this result is consistent in RNNs as well as CNNs. Another interesting direction would be to investigate which aspects of the representation present in RNNs is stable over time and which aspects vary. Additionally, in previous work [22], it was observed that fixing layers in CNNs over the course of training led to better test performance ("freeze training"). An interesting open question would be to investigate whether a similar training protocol could be adapted for RNNs.

**Acknowledgments**

We would like to thank Jascha Sohl-Dickstein for critical feedback on the manuscript, and Jason Yosinski, Jon Kleinberg, Martin Wattenberg, Neil Rabinowitz, Justin Gilmer, and Avraham Ruderman for helpful discussion.

## Footnotes

[4] This is *different* than the vector of all neuron outputs on a *single* input: $z_1(x_1), ..., z_N(x_1)$, which is also sometimes referred to as an activation vector.

[5] Including those with different topologies such that $L_1$ and $L_2$ have different sizes.

[6]We note that this is technically a pseudo-distance rather than a distance as it is non-symmetric.

[7]Details of the architectures and training procedures for this and following experiments can be found in Appendix A.4.

[8]To control for variability in CCA distance due to comparisons across representations of different sizes, a random subset of 128 filters from the final layer were used for all network comparisons. This bias should, if

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
