[Supplementary Material]

# A  Appendix

## A.1  Performance Plots for Models

We include the train/test curves for models trained in Figure 1. Comparing the curves to Figure 1, we can see that for all the models, there is a train time $t_0$ where performance is almost equivalent to final performance, but most CCA coefficients $\rho^{(i)}$ still haven't converged. This suggests that the vectors associated with these $\rho^{(i)}$ are noise in the representation, which is not necessary for doing well at the task.

**a**

**b**

**c**

Figure A1: **Performance convergence for CIFAR-10 CNNs, and PTB and WikiText-2 RNNs**.

## A.2  Additional reduction methods for CCA

**Bartlett's Test**   Another potential method to reduce across CCA vectors of varying importnace is to estimate the number of important CCA vectors $k$, and perform an average over this. A statistical hypothesis test, proposed by Bartlett [3], and known as Bartlett's test, attempts to identify the number of statistically significant canonical correlations. Key to the test is the computation of Bartlett's statistic:

$$T_k = - \left( n - k - \frac{1}{2}(a + b + 1) + \sum_{i=1}^{k} \frac{1}{(\rho^{(i)})^2} \right) \log \left( \prod_{i=k+1}^{c} (1 - (\rho^{(i)})^2) \right)$$

where, in the same notation as previously, $n$ is the number of datapoints, and $a, b$ are the number of neurons in $L_1, L_2$, with $c = \min(a, b)$. The null hypothesis $H_0$ is that there are $k$ statistically significant canonical correlations with the remaining $\rho^{(i)}$ are generated randomly via a normal distribution [3]. Under the null, the distribution of $T_k$ becomes chi-squared with $(a - k)(b - k)$ degrees of freedom. We can then compute the value of $T_k$ and determine if $H_0$ satisfactorily explains the data.

However, the iterative nature of this metric makes it expensive to compute. We therefore focus on projection weighting in this work, and leave further exploration of Bartlett's test for a future study.

## A.3  Representation Dynamics in RNNs Through Sequence (Time) Steps

Here, we investigate the utility of CCA for analyzing representations of RNNs unrolled across sequence time steps. As a toy example of CCA's benefit in this case, we first initialize a linear vanilla RNN with a unitary recurrent matrix (such that it simply rotates the hidden representation on each timestep). We then use cosine distance, Euclidean distance, and CCA to compare the hidden representation at each timestep to the representation at the final timestep (Figure A2a-c). While both cosine and Euclidean distance fail to realize the similarity between timesteps, CCA, because of its invariance to linear transforms, immediately recognizes that the representations at all timesteps are linearly equivalent.

However, as linear networks are limited in their representational capabilities, we next examine a toy case of a network involving both a linear and non-linear component. We again initialize a simple RNN with the following update rule:

$$h_{t+1} = W_{rot}h_t + \alpha \cdot \sigma(W_{rand}h_t) + b$$

**Figure A2: Toy RNN examples demonstrating that CCA is comparatively rotation invariant.** In a toy example, vanilla RNNs were initialized with a random rotation matrix and run 1000 times with a random starting hidden state and no inputs. Hidden states at each timepoint were compared to the final hidden state using cosine distance (**a**, **d**), Euclidean distance (**b**, **e**), and CCA (**c**, **f**). Due to its rotation invariance, CCA recognized all states as similar in both linear RNNs (**a-c**), and a blended linear/non-linear case (**d-f**; $h_{t+1} = W_{rot}h_t + \alpha \cdot \sigma(W_{rand}h_t) + b$, where $W_rot$ is a random rotation matrix, $W_{rand} \sim \mathcal{N}(0, I)$), while both cosine and Euclidean distance largely fail. Error bars represent mean $\pm$ std.

where $h_t$ is the hidden state at time $t$, $\sigma$ represents the sigmoid nonlinearity, $W_{rot}$ is a random rotation matrix, $W_{rand} \sim \mathcal{N}(0, I)$), and $\alpha$ is a scale factor between the linear and non-linear components. For values of $\alpha$ as high as 100 (suggesting that the nonlinear component has 100 times the magnitude of the linear component), we again find that, in contrast to CCA, cosine and Euclidean distance fail to recognize the similarity between timesteps (Figure A2d-f).

However, both of the above cases are toy examples. We next analyze the application of CCA to the more realistic situation of LSTM networks trained on PTB and WikiText-2. To do this, we unroll the RNN for 20 sequence steps, and collect the activations of each neuron in the hidden state over the appropriate sequence tokens for each of the 20 timesteps. More precisely, we can represent our output by a matrix $O$ with dimensions $(N, m)$ where $N$ is the number of neurons and $m$ is the total sequence length. Our per sequence step matrices would then be $O_0, ..., O_{19}$, with $O_j$ consisting of all the outputs corresponding to sequence tokens with index equal to $j$ modulo 20, and our matrix would have dimensions $(N, m/20)$. We can then compare $O_j$ to $O_{19}$ analogous with the comparison to the final timestep. We then apply CCA, Cosine and Euclidean distance as above. To our surprise, the hidden state varies significantly from sequence timestep to sequence timestep, Figure A4.

**Figure A3: Hidden states are nonlinearly variable over sequence timesteps.** Using CCA (left), cosine distance (middle), and Euclidean distance (right), we measured the distance between representations at sequence timestep $t$ and the final sequence timestep $T$. Interestingly, even CCA failed to find similarity until late in the sequence, suggesting that the hidden state varies nonlinearly in the presence of unique inputs.

The above result demonstrates that the hidden state varies nonlinearly in the presence of unique inputs. However, this nonlinearity could be caused by the recurrent dynamics or novel inputs. To disambiguate these two cases, we asked how the hidden state changes when the same input is re-

peated. We therefore repeat the same input for 20 timesteps, beginning the repetition after some percentage of previous steps containing unique inputs (e.g., $1\%, 10\%, ...$ through the $m$ input sequence tokens). When the repeating inputs were presented early in the sequence, CCA recognized that the hidden state was highly similar, while cosine and Euclidean distance remained insensitive to this similarity (Figure A4, light blue lines). This result appears to suggest that the recurrent dynamics are approximately linear in nature.

However, when the same set of repeating inputs was presented late in the sequence (Figure A4, dark blue lines), we found that the CCA distance increased markedly, suggesting that the nonlinearity of the recurrent dynamics depends not only on the (fixed) recurrent matrix, but also on the sequence history of the network.

**Figure A4: Hidden states vary linearly in the presence of repeated inputs.** To test whether the nonlinearity in the hidden state over sequence timesteps was due to input variability or recurrent dynamics, we measured the CCA distance (left), cosine distance (middle), and Euclidean distance (right) between sequence timestep $t$ and the final sequence timestep $T$ in the presence of repeating inputs. Interestingly, we found that when the repetition started after only a small set of unique inputs have been presented (light blue lines), CCA was able to recognize that the hidden states at each sequence timestep were highly similar. However, after many unique inputs had been delivered, the CCA distance markedly increased, suggesting that the nonlinearity of the recurrent dynamics is dependent on the network's history.

## A.4 Experimental details

**CIFAR-10 ConvNet Architecture:** The convolutional networks trained on CIFAR-10 were identical to those used in [21]. All CIFAR-10 networks were trained for 100 epochs using the Adam optimizer with default parameters, unless otherwise specified (learning rate: 0.001, beta1: 0.9, beta2: 0.999). Default layer sizes were: 64, 64, 128, 128, 128, 256, 256, 256, 512, 512, 512, with strides of 1, 1, 2, 1, 1, 2, 1, 1, 2, 1, 1, respectively. All kernels were 3x3 and a batch size of 32. Batch normalization layers were present after each convolutional layer. For the experiments in Section 3.2, all layers were scaled equally by a constant factor $\in$ 0.25, 0.5, 0.75, 1.0, 1.25, 1.5, 1.75, 2.0, 3.0, 4.0, 5.0, 6.0, 7.0.

**RNN Experiments:** RNN experiments on PTB and WikiText2 followed the experimental setup in [18] and [19]. In particular, we used the open sourced model code[9] for training the word level Penn TreeBank and WikiText-2 LSTM models, (without finetuning or continuous cache pointer augmentation). All hyperparameters were left unmodified, so experiments can be reproduced by training LSTM models using the command to run *main.py*, and then applying CCA to the hidden states, via the open source implementation[10].

**Toy Experiments:** Generate $k$ vectors in $\mathbb{R}^{2000}$ of 'signal' (iid standard normal), for $k \in 20, 50, 70, 80, 100, 120, 140, 160, 180, 199$ and concatenate this $\mathbb{R}^{k \times 2000}$ matrix with a noise matrix: $\mathbb{R}^{(200-k) \times 2000} \sim \mathcal{N}(0, 0.1)$ to. (Note that the noise being lower magnitude than the signal is something that we see in typical neural networks – work on network compression has showing that pruning low magnitude weights is an effective compression strategy.) Putting together gives matrix $X$, 200 (neurons) by 2000 (datapoints). Apply a randomly sampled orthonormal transform to the $k$ by 2000 subset of $X$ to get a new $k$ by 2000 matrix, and again add iid noise of dimensions

$(200 - k)$ by 2000 to get matrix $Y$. Apply CCA based methods to detect similarity between $X, Y$. Of particular interest are cases $k << 200$ (low dim. signal in noise).

## A.5    Additional control experiments

**Figure A5: Cosine and Euclidean distance do not reveal the difference in converged solutions between groups of generalizing and memorizing networks.** Groups of 5 networks were trained on CIFAR-10 with either true labels (generalizing) or random labels (memorizing). The pairwise cosine (left) and eucldean (right) distance was then compared among generalizing networks, memorizing networks, and between generalizing and memorizing networks (inter) for each layer. While its invariance to linear transforms enabled CCA distance to reveal a difference between groups generalizing and memorizing networks in later layers (Figure 3), cosine and Euclidean distance fail to detect this difference. Error bars represent mean $\pm$ std distance across pairwise comparisons.

**Figure A6: Cosine and Euclidean distance do not reveal the relationship between network size and similarity of converged solutions.** Groups of 5 networks with different random initializations were trained on CIFAR-10. Each group contained filter sizes of $\lambda[64, 64, 128, 128, 128, 256, 256, 256, 512, 512, 512]$ with $\lambda \in \{0.25, 0.5, 0.75, 1.0, 1.25, 1.5, 1.75, 2.0, 3.0, 4.0, 5.0, 6.0, 7.0\}$. Pairwise cosine (left) and Euclidean (right) distance was computed for each group of networks. While CCA distance revealed that larger networks converge to more similar solutions (Figure 4), cosine and Euclidean distance fail to find this relationship. Error bars represent mean $\pm$ std distance across pairwise comparisons.

**Figure A7: Relationship between network size and similarity of converged solutions is not present at initialization.** Activations at initialization (random weights) and after training (learned weights) were extracted from groups of 5 networks with different random initializations from CIFAR-10 data. Each group contained filter sizes of $\lambda[64, 64, 128, 128, 128, 256, 256, 256, 512, 512, 512]$ with $\lambda \in \{0.25, 0.5, 0.75, 1.0, 1.25, 1.5, 1.75, 2.0, 3.0, 4.0, 5.0, 6.0, 7.0\}$. While CCA distance decreases substantially for trained networks (from approximately 0.47 to 0.28), CCA distance only decreased moderately (from approximately 0.67 to 0.63) and plateaued past approximately 1000 filters. Error bars represent mean $\pm$ std distance across pairwise comparisons.

**Figure A8: Controls for RNN learning dynamics with cosine and Euclidean distance** To test whether layers converge to their final representation over the course of training with a particular structure, we compared each layer's representation over the course of training to its final representation using cosine (**a**, **c**, **e**) and Euclidean distance (**b**, **d**, **f**). In shallow RNNs trained on PTB (**a-b**), and WikiText-2 (**c-d**), both cosine and Euclidean distance display properties of bottom-up convergence, albeit with substantially more noise than CCA (6). In deeper RNNs trained on WikiText-2, we observed a similar pattern (**e-f**).

**Figure A9: Unweighted CCA and SVCCA also finds that generalizing networks converge to more similar solutions than memorizing networks, but misses several key features.** While weighted CCA (Figure 3), unweighted CCA (**a**), and SVCCA (**b**) reveal the same broad pattern across generalizing and memorizing networks, unweighted CCA and SVCCA miss several key features. First, unweighted CCA misses the fact that generalizing networks become more similar to one another in the final two layers. Second, both unweighted CCA and SVCCA overestimate the distance between networks in early layers. Error bars represent mean $\pm$ std unweighted mean CCA and unweighted mean SVCCA distance across pairwise comparisons.

**Figure A10: On test data, generalizing networks converge to similar solutions at the softmax, but memorizing networks do not.** Groups of 5 networks were trained on CIFAR-10 with either true labels (generalizing) or random labels (memorizing). The pairwise CCA distance was then compared within each group and between generalizing and memorizing networks (inter) for each layer, based on the test data. At the softmax, sets of generalizing networks converged to similar (though not identical) solutions, but memorizing networks did not, reflecting the diverse strategies used by memorizing networks to memorize the training data. Error bars represent mean $\pm$ std weighted mean CCA distance across pairwise comparisons.

## Footnotes

[9]https://github.com/salesforce/awd-lstm-lm

[10]https://github.com/google/svcca/