[Reviews · NeurIPS 2018]

Reviewer 1



EDIT: The authors response addresses my concerns about reproducibility and the appendix they promise should contain the missing details of their setup. Summary and relation to previous work: This paper presents projection weighted canonical correlation analysis (CCA) as a method to interrogate neural network representations. It is a direct continuation of previous work on singular vector canonical correlation analysis (SVCCA), addressing several limitations of SVCCA and applying the method to new models and questions. They observe that some CCA components stabilize early in training and remain stable (signal) while other components continue to vary even after performance has converged (noise). This motivates replacing the mean canonical correlation coefficient (used as a similarity metric in SVCCA) with a weighted mean, where components that better explain the activation patterns receive a higher weight. The new method is then applied in several different scenarios resulting in a number of interesting insights: when analysing converged solutions across different random initializations, mean CCA distance reveals five subgroups of solutions that map well to the clusters of solutions based on robustness to ablation found in Morcos et al 2018. networks which generalize converge to more similar solutions than those which memorize wider networks converge to more similar solutions than narrow networks RNNs exhibit bottom-up convergence over the course of training (the same result was found for CNNs in the SVCCA paper) across sequence timesteps, RNN representations vary nonlinearly Quality: In general, the claims are well-supported by rigorous experiments involving several architectures and datasets. I am thoroughly convinced of the usefulness of projection weighted CCA. The relationship between CCA vector stability over training, vector importance and generalization is not explored and would be interesting for a follow up paper. Clarity: The submission is very clearly written and easy to follow. Main results and contributions are clearly stated. However, specific details about the architectures tested and how they are trained is omitted making it impossible to exactly replicate the experiments. Originality: Projection weighted CCA is a small, simple but important improvement over SVCCA. Related work is discussed well. Significance: Others are highly likely to use projection weighted CCA and/or build on the unique experimental observations presented in this submission. Although only slightly different from the original SVCCA, researchers should use this improved version or risk not capturing important aspects of network/layer similarity. The experimental observations in particular are likely to inspire new experiments or new learning methods. Minor edits: - lines 141 and 142 repeat the same statement (that the randomization of labels was the same for all networks) - several references only include author, title and year, without any publication name, conference or arxiv ID. For example reference 8 was published at ICLR 2016 but your entry says "October 2015" References: Ari S. Morcos, David G.T. Barrett, Neil C. Rabinowitz, and Matthew Botvinick. On theimportance of single directions for generalization. In International Conference on LearningRepresentations, 2018 Maithra Raghu, Justin Gilmer, Jason Yosinski, and Jascha Sohl-Dickstein. Svcca: Singular vector canonical correlation analysis for deep learning dynamics and interpretability. In Advances in Neural Information Processing Systems, 2017.

Reviewer 2



The paper deals with the problem of comparing different neural network representations, proposing projection weighted CCA (Canonical Correlation Analysis) which builds on top of another framework [22]. Both CNN and RNN are taken into account. The improvement wrt [22] is this latter one does not effectively distinguish between the signal and the noise in the representation. Section 2-2.1 give some preliminary notions, stating essentially that CCA directions are orthogonal, but I appreciated the view of CCA w.r.t. neural network representations. Section 2.2 details why the proposed method is better than [22]: essentially, whereas [22] assumes that all CCA vectors are equally important to the representations, PWCCA proposes the projection weighting technique, which assign a weight to the single directions given by the sorted CCA coefficients. Results are of different kinds, built upon CIFAR 10 for cnn and Penn Treebank and WikiText-2 language modelling tasks for rnn: -networks which generalize converge to more similar solutions than those which memorize -wider networks converge to more similar solutions than narrow networks -across otherwise identical networks with different random initializations, networks converge to diverse clusters of solutions (Section 3.3). -RNN exhibit bottom-up convergence over the course of training -Across sequence timesteps, RNN representations vary nonlinearly My question is whether these conclusions could have been reached with [22] too, and how the cons of [22] would be bad for this kind of analysis.

Reviewer 3



Edit after rebuttal: I thank the authors for addressing my concerns and provide further details, and I updated my overall score to vote that this paper should be accepted. Summary: ======= The authors present an analysis of representations learned by neural networks using Canonical Correlation Analysis, building and expanding on the SVCCA paper[1]. I think the paper offered some interesting insights, that are likely of interest to the wider community, thus I am generally in favor of accepting this submission. But I am unsure about the clarity of the paper. Should these be addressed in the rebuttal, I am happy to update my score. The authors first introduce "projection weighted CCA", their work-horse for the analysis. In my opinion, this is the weakest point of the paper. The authors often refer to [1] as the paper they build on, whose authors used SVD on the representation vectors to get rid of noise directions, and then used CCA to align representations of different networks. The authors here omitted the pruning-noise-via-SVD step, and instead weight the projection-directions to reduce noise. It is unclear to me why this change in methodology was needed -- if noise-directions within each layer's activation are reduced via SVD, then maybe it would not be necessary to down-weight uninformative CCA directions. I suggest the author expand on how their method of analysis differs from [1], and why they needed these changes. My 2nd issue with the paper is that it gives very little details on experiments. It is impossible to reproduce the experiments in this paper, since there are too many details omitted. The authors do not specify the arcitectures of the networks they used, which would be crucial to better put the results into context. Furthermore, the experiments that lead to figure 2 are only rudimentarily described. I strongly suggest the authors add more details to make experiments reproducible (in the appendix, if space is an issue). I would also encourage the authors to provide the source code of their experiments to improve reproducability (though of course this is not a must). Minor comments: ============== - Section 3.1 speculates that "solutions found by memorizing networks were as diverse as those found across entirely different dataset labelings". It would be interesting (and presumably fairly easy for the authors) to test this instead of just speculating about it. - I'm not convinced about Figure 4b: On the one hand, larger networks are more similar (fig 4a), on the other hand, larger networks achieve better accuracy. Isn't figure 4b simply a result of these two facts? I.e., there is a latent variable "network size" that influences both performance and CCA distance. One would need to eliminate the dependance on this latent variable to see how good the connection between CCA distance & performance is. It would be more convincing if more networks of the same size (but different test accuracy) would be plotted, instead of varying the size. - The authors often use terms which are formally introduced only much later. E.g. the term "CCA vectors" is defined at line 113, but is first used from line 87 onwards. - Equation (**) should be offset from the main text so it's easier to see (the authors later refer back to this equation, and it's hard to find ) - In the 3rd paragraph of Section 3.1, the fact that "the same set of random labels was used for all networks" is repeated in the next sentence as "the randomization of CIFAR 10 labels was consistent for all networks". Why is this mentioned twice right after each other. Do these mean different things? References: ========== [1] Raghu et al, "SVCCA: Singular Vector Canonical Correlation Analysis for Deep Learning Dynamics and Interpretability", NIPS 2017